# Biparental Inheritance and Instability of kDNA in Experimental Hybrids of *Trypanosoma cruzi*: A Proposal for a Mechanism

**DOI:** 10.3390/biology14101394

**Published:** 2025-10-11

**Authors:** Nicolás Tomasini, Tatiana Ponce, Fanny Rusman, Soledad Hodi, Noelia Floridia-Yapur, Anahí Guadalupe Díaz, Juan José Aguirre, Gabriel Machado Matos, Björn Andersson, Michael D. Lewis, Patricio Diosque

**Affiliations:** 1Instituto de Patología Experimental “Dr. Miguel Ángel Basombrío” Universidad Nacional de Salta—CONICET, Salta CP4400, Argentina; 2Department of Cell and Molecular Biology, Karolinska Institute, SE-171 77 Stockholm, Sweden; 3Division of Biomedical Sciences, Warwick Medical School, University of Warwick, Coventry CV4 7AL, UK

**Keywords:** *Trypanosoma cruzi*, laboratory hybrid, kDNA inheritance, minicircle, maxicircle, Replicative MIXing (REMIX) model, kinetoplast

## Abstract

**Simple Summary:**

Chagas disease is caused by the parasite *Trypanosoma cruzi*, which carries a dense bundle of thousands of interlocked DNA rings called the kinetoplast in its mitochondrion. How this genetic material is passed on when two parasites form a hybrid has remained unclear. Here, we analyzed laboratory hybrids of *T. cruzi* and tracked the parental origin and quantity of the small DNA rings (“minicircles”) and the large DNA rings (“maxicircles”) across many generations. We found that minicircles from both parents persist together in the same offspring lines for hundreds of generations, while maxicircles persist only from one parent. We also observed that hybrids often carry increased amounts of this kinetoplast DNA compared with their parents, and that these levels vary over time. To explain these patterns, we propose a “replicative mixing” mechanism: after mating, the two parental kinetoplasts replicate, minicircles intermingle across them, and later the networks separate so that each daughter cell retains maxicircles from one parent and minicircles from both.

**Abstract:**

The mitochondrial DNA of trypanosomatid parasites consists of thousands of catenated minicircles and dozens of maxicircles that form a complex network structure, the kinetoplast (kDNA). Although kDNA replication and segregation during mitotic division are well studied, its inheritance during genetic exchange events remains unclear. In *Trypanosoma brucei*, hybrids inherit minicircles biparentally but retain maxicircles from a single parent. Although biparental inheritance of minicircles has been described in natural *Trypanosoma cruzi* hybrids, this process has not been explored in laboratory-generated hybrids of this parasite. In the present study, we analyzed kDNA inheritance in *T. cruzi* experimental hybrids using a comprehensive minicircle hypervariable region (mHVR) database and genome sequencing data. Our findings revealed biparental inheritance of minicircles, with hybrid lines retaining mHVRs from both parents for over 800 generations. In contrast, maxicircles were exclusively inherited from one parent. Unexpectedly, we observed an increase in kDNA content in hybrids, affecting both minicircles and maxicircles, and exhibiting instability over time. To explain these findings, we propose a Replicative Mixing (REMIX) model, where the hybrid inherits one kinetoplast from each parent and they are replicated allowing minicircle mixing. Instead maxicircle networks remain physically separated, leading to uniparental fixation after segregation in the first cell division of the hybrid. This model challenges previous assumptions regarding kDNA inheritance and provides a new framework for understanding kinetoplast dynamics in hybrid trypanosomes.

## 1. Introduction

The mitochondrial DNA of trypanosomatids is composed of thousands of DNA minicircles and dozens of DNA maxicircles. Such circular DNAs are catenated in a complex network, the kinetoplast (kDNA), and located within the single large mitochondrion [1]. The maxicircles encode genes necessary for mitochondrial function, such as components of the respiratory chain and ribosomal RNA genes, and some protein-coding genes, which require some post-transcriptional editing to become functional [2,3]. The high-copy-number minicircles contain hypervariable regions (mHVRs) that encode guide RNAs involved in the editing of maxicircle-encoded transcripts [4,5]. The precise organization and replication of the kDNA, which is structurally connected to the flagellum, are fundamental for trypanosomatids to adapt to different environments and successfully transmit to different hosts [6,7,8,9].

Much work has been performed to elucidate how the kinetoplast is replicated and segregated during cell division, mainly in *Trypanosoma brucei* [1,10,11,12]. Briefly, replication occurs during the S-phase of the cell cycle when minicircles are released from the kDNA network and replicated outside of it. Each released minicircle replicates via a theta mechanism [13] and is subsequently reattached to the network at specific regions known as antipodal sites (APS), which are located at opposite edges of the kinetoplast disk [10]. In contrast to minicircles, the maxicircles replicate in situ without being released, remaining stably catenated within the network [14,15]. Following replication, a final decatenation step allows for the division of the kinetoplast, which is closely linked to the segregation of the basal body of the flagellum, ensuring that each daughter cell inherits a fully functional kinetoplast-flagellum unit [16]. The basal body is crucial for flagellum organization and serves as an anchor for the kinetoplast via the Tripartite Attachment Complex (TAC) [16,17]. The duplication and movement of the basal body help coordinate the distribution of the kDNA network to the daughter cells.

The mechanism of kinetoplast inheritance during sexual exchange—where the parasite must face several challenges—remains mostly unknown. Initially described as uniparental, kinetoplast inheritance was later shown to be biparental in T. brucei hybrids, where minicircles from both parents are retained in the hybrid progeny, while maxicircles from only one parent persist over time [18,19,20]. Two mechanisms have been proposed to explain how hybrids transition from having two different kinetoplasts to a single one with mixed minicircles, but maxicircles derived from only one parent. The network fusion model proposes that the kDNA networks from both parents undergo fusion. Over subsequent generations, the low-copy-number maxicircles from one parent are lost through genetic drift, while high-copy-number minicircles from both parents persist for many more generations [20]. Alternatively, the minicircle exchange model proposes that minicircles are released from both networks without being replicated and exchanged between kinetoplasts, while maxicircles are retained in each parental network. In this scenario, kinetoplasts containing mixed minicircles, but maxicircles from a single parent are segregated between the two daughter cells, bypassing the need for genetic drift to explain parental maxicircle fixation [1]. Although much less studied, hybrids of *Trypanosoma cruzi* have been observed in both natural populations [4,21,22] and laboratory settings [23]. In the first report of experimental hybridization, Gaunt et al. [23], described six hybrid clones resulting from a cross between two TcI lines and proposed uniparental inheritance of kDNA based only on maxicircle markers. However, more recent evidence indicates that hybrid DTUs such as TcV and TcVI [24], as well as natural TcI hybrid lines [25], may exhibit biparental inheritance of minicircles. Notably, the original hybrid clones reported by Gaunt et al. were subsequently maintained in long-term in vitro culture for approximately 800 generations, allowing the detection of genomic changes through whole-genome sequencing [26].

Although hybridization in trypanosomatids has been repeatedly observed and inferred, most conclusions about kinetoplast inheritance rely on maxicircle markers, which cannot reveal whether minicircles mix, replicate, or change in dosage after hybrid formation. Moreover, no prior study has jointly quantified minicircle and maxicircle copy numbers across many asexual generations to distinguish between models of kDNA inheritance. The experimental *T. cruzi* hybrids originally described by Gaunt and colleagues offer a unique opportunity: derived from known parents and propagated for ~800 generations [26], they enable the reconstruction of minicircle repertoires and copy-number estimates. Reanalyzing these hybrids with a curated minicircle hypervariable region (mHVR) database and complementary PI-based cytometry constraints, we set out to (i) trace parental contributions to the minicircle pool over time, (ii) quantify total kDNA dosage for minicircles and maxicircles, and (iii) discriminate among alternative inheritance mechanisms and their predictions—thereby supporting a revised model of kDNA inheritance in hybrid trypanosomes.

## 2. Materials and Methods

### 2.1. Data Source

Raw genomic sequencing data from parental and hybrid lines of T. cruzi were obtained from the NCBI Sequence Read Archive (SRA) (accession: PRJNA748998). These hybrids, originally described by Gaunt et al. [23], resulted from a cross between two TcI lines and were maintained in long-term in vitro culture, allowing the detection of genomic changes over approximately 800 generations [26] (Appendix A). The sequence data used in this study were obtained from parental lines sequenced approximately 70 generations after hybridization, and from hybrid lines sequenced at around 95 generations. In addition, sequence data were obtained in parental and hybrid lines again after approximately 800 generations of in vitro culture, as detailed in Matos et al. [26] (Appendix A). Propidium iodide (PI)–based total DNA content profiles from epimastigote cultures of parental lines and hybrids at different time points were obtained from Matos et al. [26].

### 2.2. Sequence Data Processing and mHVR Frequency Estimation

Raw reads were trimmed and quality-filtered using Trimmomatic v0.39 [27], with parameters LEADING:30 TRAILING:30 SLIDINGWINDOW:4:20 MINLEN:40 to remove low-quality bases and adapter sequences. Processed reads were then mapped to a reference database of minicircle hypervariable regions (mHVRs) [28] using BWA-MEM v0.17 [29] with default parameters. The reference database corresponds to the 95% identity mHVR dataset, in which sequences were clustered based on ≥95% pairwise nucleotide identity. Each cluster of sequences was considered a mHVR class. Within each cluster, a single representative sequence—the most frequently observed—was selected to construct a reference set. The resulting database is available at: https://raw.githubusercontent.com/ntomasini/cruzityping/refs/heads/v1.0.1/rep_set95_FINAL.fasta (accessed on 17 May 2025). After mapping reads against the mHVR database, a custom python v3.11 script was used to retain only those with a read depth greater than 10× over at least 170 base pairs for further analysis. Subsequently, the average read depth was calculated for each representative mHVR to estimate its class abundance and build a frequency table as in [28]. Shared mHVRs classes between parental lines were defined as those with an average read depth greater than 50× in both lines, while unique mHVRs were defined as those with average read depth less than 5× in one parent and greater than 50× in the other one. These parameters were selected to reduce the noise caused by low frequency or artifactual mHVRs.

### 2.3. Variant Calling in Maxicircles

Trimmed reads were aligned using BWA-MEM to the maxicircle reference sequence of the *Trypanosoma cruzi* Sylvio X10 strain (GenBank: FJ203996.1), a representative of DTU TcI. Bam files were sorted and indexed with SAMtools v1.21 [30]. To assess coverage, we used the samtools depth function, and positions with <20 aligned reads were defined as low-coverage regions. These intervals were converted into a BED file via a custom AWK script. Next, we applied BCFtools v1.21, particularly, bcftools mpileup and bcftools call to identify multi-allelic variants (option -m) in regions with sufficient read support according to the coverage BED file previously generated. Finally, a consensus sequence was built with bcftools consensus, masking low-coverage positions as ‘N’ based on the BED intervals. This approach yielded a reference-guided assembly in which poorly supported regions were not inferred from the reference genome. For each polymorphic position, read mapping was visually inspected using IGV [31] to detect heteroplasmy with a frequency higher than 1% of the alternate allele.

### 2.4. Simulation Model for Maxicircle Drift

We implemented a discrete-generation Monte Carlo model to evaluate the suitability of the network fusion model to fix maxicircles from one parent as observed in the hybrids. Each parasite was represented by a “kinetoplast” containing a fixed number of maxicircles (labeled as “A” or “B,” corresponding to parental origin), and the population evolved for 95 generations (the estimated number of generations since hybridization in the experimental hybrids analyzed here). In every generation, maxicircles were duplicated, randomly shuffled, and split into two sets, each inherited by one of the two progeny, according to the model of maxicircle replication during parasite division [18]. If this procedure yielded more parasites than a specified maximum population size (set to 10,000 parasites), a random subset of 10,000 parasites was chosen to remain in the population. In each generation, the number of kinetoplasts that had exclusively “A” or “B” maxicircles (indicating fixation) versus those that retained both types (not fixed) were recorded. With each combination ten replicates were generated to capture stochastic variability. This simulation script can be run in Google Colab cloud by accessing a GitHub repository (https://github.com/ntomasini/MaxiDrift).

### 2.5. kDNA Quantification by Read Depth

To quantify the kDNA content, we calculated the read depth of mHVRs, conserved regions of minicircles, and maxicircles. This was normalized by the average read depth of disomic, trisomic or tetrasomic regions of chromosome 1 (positions 124,001–613,000) and 18 (positions 298,001–498,000), from the Sylvio X10 reference genome downloaded from TriTrypDB (https://tritrypdb.org/tritrypdb/app/record/genomic-sequence/TcX10_chr1 and https://tritrypdb.org/tritrypdb/app/record/genomic-sequence/TcX10_chr18 both accessed on 17 May 2025). These regions were selected by visual inspection of read depth because of their homogeneous values. Copy number of these nuclear regions were assessed by read depth and variant calling analysis after mapping trimmed genome reads against it using BWA-MEM. Each BAM file was then sorted with Samtools and processed with MarkDuplicates (Picard) from GATK software [32] to identify and mark duplicate reads. The reference genome was indexed with samtools faidx, and a sequence dictionary was created using CreateSequenceDictionary (Picard) from GATK to ensure compatibility with the variant caller. We used GATK HaplotypeCaller with—standard-min-confidence-threshold-for-calling 10 and—min-base-quality-score 20 for SNP detection, producing a single VCF file per sample. After variant calling, bcftools filter was applied to each resulting VCF (MQ > 50) to remove variants with lower mapping quality. The alternate allele frequencies (AAF) were addressed for patterns indicating disomy (AAF = ~0.5), trisomy (AAF = ~0.33 and ~0.66) and tetrasomy (AAF = ~0.25, ~0.5 and ~0.75). The average read depth of chromosome 1 and 18 regions (Appendix A) was used to normalize read depth from maxicircles, the conserved region of minicircles and mHVRs.

Read depth for the minicircle conserved region was estimated by mapping the genome reads against the conserved region of a minicircle (GenBank: X04680.1:1104-1223) using BWA-MEM. To calculate minicircle abundance, the median read depth was divided by four—the number of conserved regions per minicircle—and normalized to the nuclear read depth (as described above), yielding the average number of minicircles per parasite. The standard deviation was calculated based on variations in read depth across the conserved region. Alternatively, the total minicircle number was estimated from the sum of all mHVR average depths, using the same normalization strategy applied to the conserved region. A bootstrap method with 500 replicates was used for estimation of the standard deviation. The number of maxicircles per parasite was calculated with a similar approach by using the previous mapped reads for variant calling; however, instead of the median, the 75th percentile of read depth was used due to variable coverage across the conserved region of the maxicircle. Reads from AT-rich segments were particularly underrepresented (Appendix A), so adopting the 75th percentile avoided the underestimation of the copy number due to low sequencing coverage of AT-rich regions.

### 2.6. kDNA Quantification from PI Flow Cytometry and Nuclear Read Depth

We independently estimated kDNA content from flow–cytometry data to mitigate potential under- or overestimation caused by biased DNA purification, given that kDNA has distinct physical properties from nuclear DNA (i.e., interlocked DNA rings) [26]. In this approach, kDNA content in the hybrid was inferred as the difference between the total DNA measured by PI fluorescence and the nuclear DNA inferred from sequencing. This approach avoids estimating kDNA quantities based on kDNA read depth.

Let $N_{p}$ be the parental nuclear DNA, $K_{p}$ the parental kDNA, $N_{h}$ the hybrid nuclear DNA, and $K_{h}$ the hybrid kDNA. The total DNA ratio can be defined as$$R\;=\;\frac{N_{h}+K_{h}}{N_{p}+K_{p}}$$
and it was measured by flow cytometry (PI fluorescence) on hybrid and parent lines processed in parallel [26].

From the definition of R,$$R\;(N_{p}+K_{p})\;=N_{h}+K_{h}\Rightarrow K_{h}=R\;(N_{p}+K_{p})\;-N_{h}$$

We inferred $N_{p}$ and $N_{h}$ from whole-genome sequencing by normalizing total mapped bases to the Sylvio reference genome to the median depth of a single-copy nuclear region with known somy as in the previous section.

Because $K_{p}$ is not identifiable from these data alone, we assume a plausible parental kDNA fraction between 0.2 and 0.3 [33,34] to estimate $K_{h}$ and we report a range for it. To estimate the number of minicircles and maxicircles from $K_{h}$ we assumed a minicircle size of 1.4 kb [35] and a maxicircle size of 50 kb [36] and the minicircle-to-maxicircle number ratio was derived from sequencing depths.

### 2.7. Estimation of Parental Contributions

To estimate the fraction of mHVRs inherited from one parental strain (Parent 1) in a hybrid, a multinomial maximum-likelihood framework was applied. For each mHVR class *i*, the expected frequency in the hybrid was modeled as$$f_{mix,i}(p)=p\cdot f_{A}(i)+(1-p)\cdot f_{B}(i),$$
where *p* represents the mixing proportion contributed by Parent 1. The log-likelihood of observing the hybrid counts *k_i_* under this model was$$LL\left(p\right)=\sum_{i}k_{i}ln\left[f_{mix,i}\left(p\right)\right]$$

We maximized this log-likelihood by numerically minimizing its negative (-*LL*) using *minimize_scalar* from the *scipy.optimize* Python library. The optimal *p* obtained corresponds to the estimate of the inherited proportion from Parent 1 in each hybrid. A similar approach was used by maximizing the Pearson correlation coefficient between the observed mHVR class frequencies in the hybrid and *f_mix_*.

## 3. Results

### 3.1. Experimental Hybrids Exhibit mHVRs from Both Parents

The composition of minicircle hypervariable regions (mHVRs) in each parasite line was estimated by mapping high-quality, filtered reads to a reference database of *T. cruzi* mHVR sequences. The mHVR composition analysis revealed significant differences between the parental and hybrid lines. A total of 3472 mHVR classes were identified in the parental lines. Both parental lines shared 2079 (59.9%) of these classes, whereas the remaining classes were specific to only one of them. At the beginning of the experiment (approximately generation 95 after hybridization), the hybrids showed a composite mHVR repertoire drawn from both parents, with <1% of classes not detected in either parent. In the hybrids, 77.1 ± 2.9% of detected classes were those already shared by the parents, and 22.9 ± 2.9% corresponded to classes unique to one parent or the other. This percentage of classes contributed by each progenitor strongly suggests biparental inheritance of the mHVRs (Figure 1A). In addition, relative mHVR abundance was also estimated by read depth on the different mHVR classes and a similar pattern was observed (Figure 1B).

The contribution of each parental strain was also analyzed. Maximization of the likelihood and Pearson’s *r* comparing the frequency of mHVR classes in the hybrids and a linear combination of the two parents revealed that mHVR abundances in the hybrids best predicted a mix of around 41–46% mHVRs from parent 1 and 54–59% from parent 2 (Table 1). This result suggests similar contribution from both parental kinetoplasts.

### 3.2. Minicircles from Both Parents Persist After 800 Generations in the Hybrids

The mHVR composition in the hybrids was also analyzed after 800 mitotic generations of in vitro culture maintenance to assess the genetic stability of hybrid kinetoplasts over time. If the minicircles from one parent were completely (or nearly completely) lost, e.g., due to a segregation mechanism of entire kinetoplasts, this would suggest that minicircles from both parents are maintained in separate networks. Specific mHVRs from both parents persisted after 800 generations (Figure 2B,C and Appendix A). This result strongly suggests that parental minicircle networks were intermixed in the hybrids. However, the frequency of each mHVR class became highly variable over time and a significant reduction in the strength of correlation of abundances between the hybrids and the parents was observed (Figure 2A vs. Figure 2B and Appendix A left vs. Appendix A right). In addition, several mHVR classes that were abundant in parents and hybrids early in the experiment (Figure 2A and Appendix A left) were completely lost after 800 generations (Figure 2B and Appendix A rigth). This result suggests the mHVR composition is unstable after hybridization and undergoes stochastic variation during subsequent clonal microevolution.

### 3.3. Hybrids Retain Maxicircles from a Single Parent

Maxicircle inheritance was also studied. Conserved regions with coverage >20× (a total of 12,544 bp) were selected for variant calling analyses. Fifteen polymorphic sites that varied between the parental lines were detected (Figure 3A). The 1D12 hybrid had maxicircles identical to those of parental line 2. In contrast, maxicircles from the 1C2 and 2C1 hybrids were identical to parental line 1, as previously reported [26]. No evidence of heteroplasmy was observed in such polymorphic sites for any hybrid. We built a simple simulation to address the random segregation of kDNA maxicircles and to evaluate the proportion of cloned lines expected to fix maxicircles from one parent. The proportion of parasites that lost maxicircles from one or the other parental line was used as an estimator of the fixation probability. This probability increased with the number of generations after hybridization and decreased with the number of maxicircles in the hybrid (Figure 3B). At ninety-five generations the probability of six fixed hybrid lines, as observed by [23], was *p* = 0.19 for 28 maxicircles in the hybrid parasite but it drops to *p* < 0.02 if the hybrid contains 44 maxicircles. These results indicate that random segregation of maxicircles is unlikely to account for parental maxicircle fixation over such a low number of generations if the number of maxicircles in the hybrid is higher than forty.

### 3.4. kDNA Content Is Increased and Unstable in Hybrids

Under the assumption of biparental inheritance of the kDNA, the network fusion model predicts that hybrid parasites would initially have approximately double the kDNA content, while the minicircle exchange model predicts that they would retain roughly the same amount as the parental lines. Coverage analyses—normalizing kDNA sequencing depth to nuclear regions with known somy—suggested that all hybrid lines contained a higher amount of both minicircles and maxicircles compared to the parental lines a few generations after hybridization (Figure 4C,E). A strong correlation (*r* = 0.89, using the read-depth estimation on the conserved region of the minicircles) between minicircle and maxicircle abundances was also observed (Figure 4B). However, the estimated number of minicircles and maxicircles in the hybrids was 3–6 times higher than in the parents (Figure 4C,E). Such an increase is unlikely according to cytometry data from the previous study where the hybrids were originally analyzed [26]. However, estimations of nuclear DNA based on read depth were consistent to those observed by PI fluorescence (Figure 4A). Consequently, we estimated kDNA content in the hybrid as the difference between the total DNA measured by PI fluorescence and the nuclear DNA content inferred from read depth. An increase in minicircles and maxicircles was also observed in the hybrids but in proportions lower than estimations based on read depth of the kDNA (Figure 4D,F). Increases were on average 1.8–2.0× for maxicircles and 1.3–1.4× for minicircles assuming kDNA corresponds to 20% or 30% of the whole DNA in the parental lines, respectively (Figure 4D,F and Appendix A).

## 4. Discussion

Our findings provide new insights into the complexity of kDNA inheritance and organization in *T. cruzi* hybrids. Using our comprehensive minicircle hypervariable region (mHVR) database [28], we traced the parental origin of mHVRs in closely related parasite lines at initial hybridization and after approximately 800 generations of clonal in vitro microevolution. We observed that hybrid lines inherited minicircles biparentally, resulting in a mixture of parental minicircles that persists even after 800 generations, despite stochastic variation in their relative abundances between hybrid lines over time. This finding suggests that minicircles become physically mixed, enabling the long-term coexistence of parental minicircle variants within a single clonal lineage. We previously proposed the biparental inheritance of minicircles in naturally occurring *T. cruzi* hybrid DTUs [24] and naturally occurring hybrid lines [25]; however, this study constitutes the first report of biparental minicircle inheritance in experimental *T. cruzi* hybrids.

In contrast, the maxicircles did not exhibit such mixing. Instead, maxicircles inherited from only one parent were quickly fixed in hybrids. Several studies have reported this observation in natural hybrids [4,22,37]. In the experimental hybrids analyzed here, network fusion followed by neutral drift could, in principle, produce maxicircle fixation if the effective per-cell copy number were very low (<30). However, that scenario would require adopting a parental kDNA fraction in the lower bound (~20% of total DNA), which—given our back-calculations—implies <15 maxicircles and ~10,000 minicircles per cell, both below canonical values for *T. cruzi* (20,000–30,000 minicircles; ~50 maxicircles) [33,34]. By contrast, using a literature-consistent kDNA fraction closer to 30% places parents at ~20–25 maxicircles and ~20,000 minicircles, and hybrids at >40 maxicircles—figures that accord with classical measurements and with our PI-based cytometry constraints. Thus, while drift cannot be excluded under vanishingly small copy numbers, the balance of evidence in these TcI-TcI hybrids favors higher effective copy numbers, making a purely drift-driven fixation unlikely. A more parsimonious interpretation is that parental maxicircle subnetworks remained largely segregated and were inherited as units. This interpretation is consistent with previous observations that maxicircles are physically interconnected in *T. brucei*, forming a distinct network embedded within the broader kinetoplast network [38]. Additionally, unlike minicircles, maxicircles do not detach from the kinetoplast network during replication [39], making it unlikely that parental maxicircle networks would physically intermix. Although recombination within kDNA has been demonstrated—particularly in *T. brucei* as part of repair pathways [40]—inter-parental maxicircle mosaicism was not observed in these hybrids because parental maxicircle subnetworks probably remain physically segregated.

Taken together, these observations argue that maxicircles behave as largely segregated units that fix rapidly from a single parent, while minicircles mix. Accordingly, any functional consequences in hybrids would more plausibly stem from changes in the gRNA repertoire than from maxicircle polymorphisms. Although precise edited mRNA sequences require direct transcript sequencing, only two parental polymorphisms fall within editing-requiring regions: (i) an A insertion at 2281 in P1 corresponds to one additional U in the pre-edited ND9 mRNA (because ND9 is encoded on the reverse strand), and (ii) a variant at position 3945 adds one U in COIII from P2. Given canonical gRNA targeting, both contexts are expected to yield identical edited products; thus, a functional impact on ND9 or COIII is unlikely. More broadly, stable biparental minicircle mixing can buffer editing against local loss of specific gRNA classes, potentially restoring editing capacity where needed [41].

An interesting consequence of hybridization was the increase in kDNA we observed, affecting minicircles and maxicircles. This pattern does not align with a pure minicircle-exchange model, which predicts compositional mixing of minicircle classes without a coordinated expansion of total kDNA dosage. Instead, the parallel rise in both components is more consistent with the early post-hybridization replication of parental kDNA subnetworks before segregation in the first mitotic division in the hybrid—yielding higher effective copy numbers than expected under exchange and segregation alone.

To explain these observations, we propose the REplicative MIXing (REMIX) model (Figure 5). According to this model, kinetoplasts—derived from different parents—likely come into proximity in the mitochondrion of the newly formed hybrid parasite cell. Minicircles detach from their original networks, undergo replication and subsequently reattach randomly to the antipodal sites of any parental kDNA network. It is important to note the difference with the minicircle exchange model, where minicircles are released but not replicated before reattachment. In contrast, we propose that minicircles undergo replication upon release, following the typical kDNA replication process. This replication and cross-attachment mechanism explains both the increase in minicircle numbers and their mixing in similar proportions, as observed in this study. However, maxicircle networks remain largely separate, replicating in a way that preserves their parental identity. Consequently, the hybrid replicates each parental network independently, leading to the formation of four kDNA networks, which then segregate into daughter cells resulting in doubled kDNA levels in each one. The data in this manuscript do not provide information about the basal body and tripartite attachment complex (TAC) dynamics during kDNA replication in the F1 hybrid. However, it is possible to hypothesize that they are not replicated in the F1 hybrid because two basal bodies are already present. This probably prevents the daughter cells of the hybrid from having more than one basal body and, consequently, more than one flagellum [42]. Consequently, the kDNA should be replicated but not segregated because basal body and TAC replication are essential for kDNA network segregation [7,43,44]. Although generating or detecting experimental F_1_ hybrid formation is challenging in both *T. cruzi* and *T. brucei*, several experiments involving targeted disruption of the TAC may provide relevant information. In *T. brucei*, the TAC is composed of multiple proteins organized into distinct structural modules. The exclusion zone filaments (EZF) contain proteins such as p197 and TAC65 [7,45], whereas the mitochondrial outer membrane anchors contain TAC60, TAC42, TAC40, and pATOM36 [7,46]. The mitochondrial inner membrane harbors p166/TAC102 [47], where the N-terminal region is associated with unilateral filaments and kDNA, and the C-terminal region interacts with TAC60 in the outer membrane. Inhibition of these proteins severely affects kDNA segregation, leading to defects in kinetoplast anchoring to the flagellar basal body. In TAC102-silenced [48] or TAC40-silenced [46] mutants, replicated kDNA networks fail to separate properly, resulting in either a single oversized kinetoplast disc or complete kDNA loss. Electron microscopy of the oversized kinetoplast showed that it contained several separated but packed kDNA networks [46].

The REMIX model requires that minicircles be released from the network, replicated, and reattached. Although there is strong evidence regarding the occurrence of these processes, mainly in *T. brucei* and *Crithidia fasciculata* [1,49,50,51,52], the exact mechanisms are unclear. Two main models have been proposed to explain minicircle replication in *T. brucei*. The classical model suggests that minicircles are first released from the kDNA network, replicated externally at the kinetoflagellar zone, and subsequently migrate to antipodal sites on opposite sides of the kinetoplast disc [1,49,51]. However, this classical view is challenged by localization studies showing that key proteins involved in minicircle replication (e.g., RBP38, helicases and primases) and early replicative intermediates are primarily found at the antipodal sites rather than throughout the kinetoflagellar zone [10,53,54,55,56]. A revised loose-diploid model proposes that minicircle release, replication, and reattachment occur predominantly in the same lobe of the disc. Under this model, incorporation of minicircles derived from one of the strands (lagging strand, replicated through Okazaki fragments and requiring additional gap repair) is delayed relative to the other strand. This staggered incorporation results in a spatial separation of the sibling minicircles and a bilobed organization of replicated minicircle sets. This organization facilitates a balanced distribution of essential minicircles during kinetoplast segregation, while still allowing for some degree of biased distribution [10]. The high variation in mHVR frequency observed after 800 generations in hybrids suggests that the loose-diploid model may also be applicable to *T. cruzi*, although further studies are needed to confirm this. Regardless of the exact site of minicircle replication, reattachment occurs at the antipodal sites (APS). Thus, the REMIX model proposed for kDNA inheritance in hybrids remains compatible with both existing models of minicircle replication.

Finally, it remains unclear whether clones with higher kDNA content can persist long-term in a natural life cycle. Particularly, the laboratory hybrids analyzed here infected SCID mice and were recovered without any substantial change in total DNA content [57]. However, it is likely that parasites with lower kDNA levels would be favored in the long term. Perturbations that enlarge or mis-segregate the kinetoplast (e.g., depletion of TAC components such as TAC102 or TAC40, or replication factors) produce oversized/unstable networks, growth impairment, and frequent kDNA loss—consistent with replication/segregation burden rather than increased editing or respiratory throughput [46,48]. Conversely, an expanded, biparental minicircle repertoire can buffer or restore RNA editing if particular gRNA classes have been lost or are limiting in one or both parents. This framework is consistent with our observation that kDNA content declines over ~800 generations despite persistent biparental minicircle ancestry, suggesting selection against sustained kDNA enlargement with a potential short-term buffering benefit when gRNA classes are limiting. In this context, analyzing kDNA levels in natural *T. cruzi* populations could provide insights into whether natural hybrids tend to exhibit increased kDNA content compared to non-hybrid lineages.

Biparental mixing of minicircles—documented in natural and successful hybrid DTUs such as TcV and TcVI [25]—reshaped the repertoire of guide RNAs [4], which in principle can buffer editing of maxicircle transcripts against loss of specific gRNA classes at the cost of a higher replication/segregation burden. This trade-off may also help explain why most laboratory hybrids show net losses of minicircles and maxicircles after ~800 generations while still maintaining mixed minicircle ancestry. Such dynamics may clarify why some hybrid lineages (e.g., TcV/TcVI) attain broad geographic ranges and epidemiological importance [58]. Practically, stable minicircle mixing explains why minicircle-based typing can retain biparental ancestry signals long after hybrid formation [25], whereas maxicircle markers read as uniparental which has relevance in epidemiological studies. Beyond kinetoplastids, biparental inheritance of mtDNA is uncommon with mechanisms to avoid it and dissecting how kinetoplastids tolerate and even exploit biparental transmission can clarify the selective pressures that led most eukaryotes to evolve strict maternal inheritance, germline bottlenecks, and active paternal mtDNA elimination [59,60,61].

## 5. Conclusions

Our work resolves key aspects of kDNA inheritance in *T. cruzi* experimental hybrids. Using mHVR tracing, we show that minicircles are inherited biparentally and remain mixed over ~800 clonal generations with strong variations in frequency, whereas maxicircles do not intermix and instead fix rapidly from a single parent. Simulations argue against maxicircle fixation by neutral drift and instead favor segregation of largely intact parental maxicircle subnetworks. Notably, hybridization was accompanied by an increase in total kDNA affecting both minicircles and maxicircles, a pattern incompatible with a pure minicircle exchange without replication view.

We therefore propose the REplicative MIXing (REMIX) model: in the nascent hybrid, minicircles detach, replicate, and reattach across parental networks, producing persistent compositional mixing and an early rise in copy number, while parental maxicircle subnetworks also replicate but remain separate and are inherited as units. This model provides a conceptual framework for understanding the interplay between minicircle and maxicircle networks, their replication, and their segregation, setting the stage for future investigations into the molecular mechanisms governing kDNA evolution and stability after hybridization in *trypanosomatids.*

## Figures and Tables

**Figure 1 biology-14-01394-f001:**
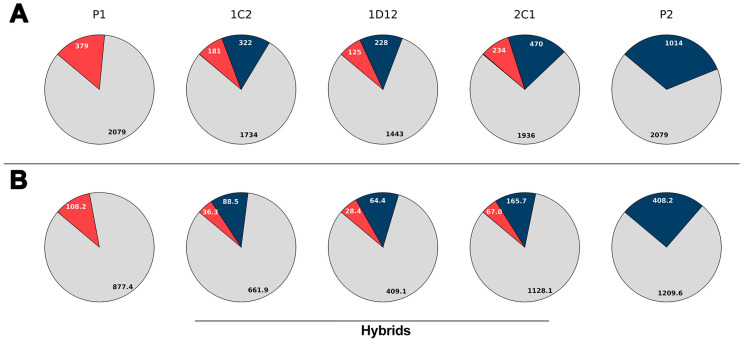
Distribution of kDNA minicircle hypervariable region (mHVR) classes and their abundances in parental (P1, P2) and hybrid (1C2, 1D12, 2C1) clonal lines of *Trypanosoma cruzi*. (**A**) The pie charts represent the proportion of mHVR classes shared with both parental strains (gray), mHVR classes specific to P1 (red) and P2 (blue). (**B**) Abundance of mHVRs in parental and hybrid *T. cruzi* lines measured as the sum of averages of sequencing depth for each mHVR class. The values are expressed in thousands (×10^3^). mHVR classes only observed in the hybrids constitute less than 0.2% and they are not shown in the graphs for clarity.

**Figure 2 biology-14-01394-f002:**
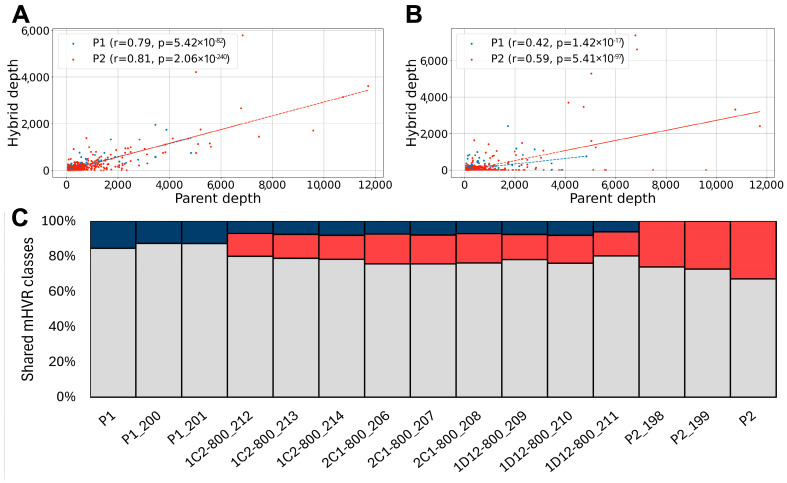
Hybrids retain mHVR from both parental lines after 800 generations. (**A**) Correlation between mHVR read depth in parent 1 (blue) or parent 2 (red) and 1C2 hybrid at first generation and (**B**) after 800 generations (SRR15686212). Only parental-specific mHVR classes were considered. Pearson’s *r* coefficient for correlation between the parental and hybrid abundances and its *p* value are shown in each graph. Note that at generation 800, *r* and p values diminished and several mHVR classes were lost in the hybrid (dots in the x-axes). (**C**) Proportion of minicircle hypervariable region (mHVR) classes in hybrid and parental cloned lines shared with each parental line. Bars represent hybrid and parental lines at 800 generations, with gray segments indicating mHVR classes detected in both parents, red segments indicating classes specific to P2, and blue segments indicating classes specific to P1. All in vitro evolved hybrids retained minicircles from both parents despite differences in copy number.

**Figure 3 biology-14-01394-f003:**
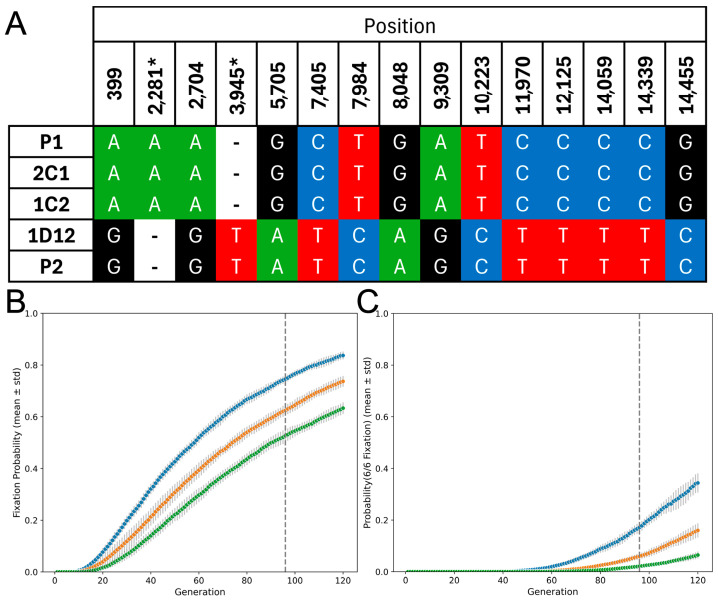
kDNA maxicircle inheritance in the hybrids. (**A**) Fifteen polymorphic sites detected in the maxicircles of parental lines (P1 and P2) and their hybrids (2C1, 1C2, 1D12). Letters denote the nucleotide present at each position, and colors highlight the distinct variants. Polymorphisms in regions that encodes mRNAs requiring editing are denoted by an asterisk (**B**) Simulation of the probability of fixing one parental maxicircle lineage over successive generations under random segregation and equal contribution from each parent. Each curve represents a different initial number of maxicircles per parasite (blue = 28, orange = 36, green = 44). The vertical dashed line marks the approximate number of generations after hybridization when the hybrids were sequenced. (**C**) Estimated probability of observing fixation of maxicircles from the same parent in all six hybrid clones, under the same random segregation model and color scheme as in (**B**). The means ± standard deviations are shown, calculated over 10 simulation runs.

**Figure 4 biology-14-01394-f004:**
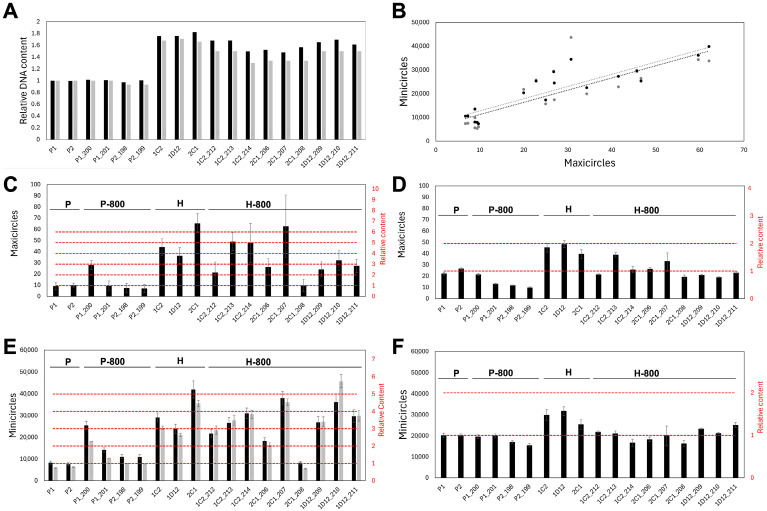
Mitochondrial DNA quantification in parental and hybrid lines after hybridization and 800 generations later. (**A**) Relative total DNA content to Parental 1 (P1) determined by flow cytometry (gray bars) and relative nuclear DNA content (black bars) estimated by the sum of read-depth across the whole reference genome and normalized to the average read depth of regions with known somy in chromosomes 1 and 18. (**B**) Scatter plot of maxicircle counts vs. minicircle counts estimated based on read depth of conserved (black dots, *r* = 0.89) or hypervariable regions (gray dots, *r* = 0.79) of the minicircles, including a linear regression (dotted lines). (**C**) Maxicircle number estimation and relative content to P1 for each parasite line based on read depth in the conserved region of the maxicircles. (**D**) Maxicircle number estimation and relative content to P1 for each parasite line based on difference between total DNA content estimated by cytometry and nuclear DNA content estimated by read depth assuming P1 kDNA corresponds to 30% of the total DNA. (**E**) Minicircle number estimation and relative content to P1 for each parasite line based on read depth in the conserved region (black bars) and hypervariable region (gray bars). (**F**) Minicircle number estimation and relative content to P1 for each parasite line based on difference between total DNA content estimated by cytometry and nuclear DNA content estimated by read depth assuming P1 kDNA corresponds to 30% of the total DNA.

**Figure 5 biology-14-01394-f005:**
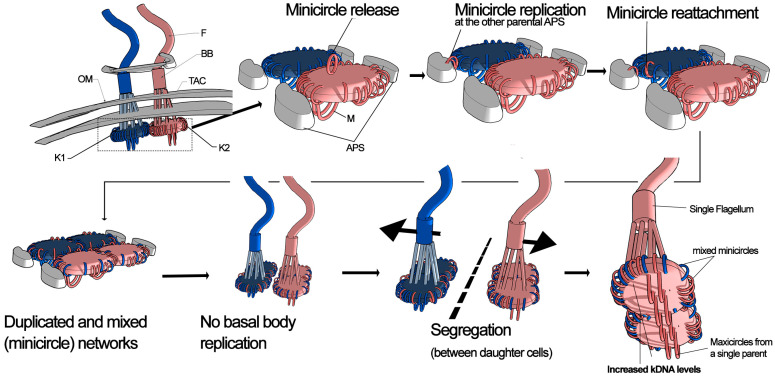
Replicative Mixing (REMIX) model. This model explains the increased and variable kDNA content observed in the hybrid lines. After hybridization, kinetoplasts from different parents (blue and red) come into proximity within the shared mitochondrial space (probably after mitochondrial fusion). Minicircles are released from their parental networks, replicate at antipodal sites (APS) of the opposing parental kinetoplast, and reattach randomly. In contrast, maxicircles remain within their original subnetwork and replicate without mixing. The resulting kinetoplast contains duplicated and mixed minicircle networks, while maxicircles retain their parental identity. In contrast to typical kDNA replication, it is not expected basal body (BB) replication and tripartite attachment complex (TAC) duplication. Pro-basal bodies not shown for clarity. The kinetoplasts from different parents finally undergo segregation resulting in increased kDNA levels but a single BB and a single flagellum per parasite. Abbreviations: F, flagellum; BB, basal body; TAC, tripartite attachment complex; OM, outer mitochondrial membrane; K1/K2, parental kinetoplasts; APS, antipodal sites; M, maxicircles.

**Table 1 biology-14-01394-t001:** Estimation of minicircle contribution of parental 1 line to different hybrids by maximum likelihood and maximization of the Pearson *r* coefficient.

Hybrid Line	Maximum Log-Likelihood	P1 Proportion (Likelihood) ^1^	Maximum Pearson’s *r*	P1 Proportion (Pearson’s *r*) ^2^
1C2	9.4 × 10^6^	0.46	0.80	0.41
2C1	1.7 × 10^7^	0.44	0.69	0.42
1D12	6.5 × 10^6^	0.44	0.69	0.44

^1^ Proportion of mHVR from parent 1 in the hybrid estimated by maximum likelihood. ^2^ Proportion of mHVR from parent 1 in the hybrid estimated by maximum Pearson *r* coefficient.

## Data Availability

All the sequences used in this study are available at NCBI Sequence Read Archive (SRA) (accession: PRJNA748998).

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
