# Peer review of "Biparental Inheritance and Instability of kDNA in Experimental Hybrids of Trypanosoma cruzi: A Proposal for a Mechanism"

_biology, 2025, doi:10.3390/biology14101394_

Round 1
Reviewer 1 Report
Comments and Suggestions for Authors
The article presents interesting data on the inheritance of kDNA (maxi- and minicircles) in T. cruzi hybrids. I would like to see the following points discussed in more detail in the manuscript:
-
Minicircles are important for the editing of maxicircle-encoded genes. What is the level of similarity between the edited genes of the parental strains? Could this have an impact on the hybrids?
-
The article proposes the hypothesis that maxicircles replicate in the hybrid, but only the DNA from one of the parental strains remains. Since recombination has already been demonstrated in T. brucei (https://doi.org/10.1016/j.dnarep.2018.11.005), why is no recombination observed between parental maxicircles?
-
It would be interesting to discuss whether a higher amount of kDNA (maxi- and minicircles) could interfere with mRNA levels, RNA editing, and mitochondrial function.
-
Why do hybrids lose kDNA after 800 generations?
Minor points
-
In the legend of Figure 2 (lines 310–312), the colors representing the parental strains were swapped.
Author Response
We are grateful for the reviewer’s careful reading and suggestions regarding minicircle function, maxicircle recombination, dosage effects, and long-term dynamics.
Comment: The article presents interesting data on the inheritance of kDNA (maxi- and minicircles) in T. cruzi hybrids. I would like to see the following points discussed in more detail in the manuscript:
- Minicircles are important for the editing of maxicircle-encoded genes. What is the level of similarity between the edited genes of the parental strains? Could this have an impact on the hybrids?
Response: We agree that the functional consequences of potential differences in edited transcripts are important. Exact sequences of fully edited mRNAs cannot be inferred with certainty without sequencing the edited transcriptome. Nevertheless, we can estimate whether parental polymorphisms fall within editing-requiring regions and whether they would be neutralized by canonical guide RNA (gRNA)–directed insertions/deletions. In our dataset, only two parental polymorphisms map to editing-requiring regions: (i) an A insertion at P1:2281 corresponds to one additional U in the pre-edited ND9 mRNA, since ND9 is encoded on the reverse strand; and (ii) a polymorphism at position 3945 implies one additional U in the P2 region of the pre-edited COIII transcript. Under the rules of U-insertion/deletion RNA editing in trypanosomatids, the same gRNA will direct the same edited product in both parental contexts; thus, we do not expect changes in the final edited ND9 or COIII mRNAs in the hybrids on the basis of these variants. More broadly, genetic exchange has been proposed as a route to restore editing capacity when specific gRNA classes are lost or compromised, an idea that is congruent with our observation of stable biparental minicircle mixing.
We marked the relevant variants in Fig 3A. Polymorphism positions were corrected to match the maxicircle reference used for variant calling. Previously, the positions reflected the site in the alignment. With these new coordinates it is possible to address if polymorphisms are located in genes whose mRNAs require editing.
We have also added the following sentences to the Discussion:
“Taken together, these observations argue that maxicircles behave as largely segregated units that fix rapidly from a single parent, while minicircles mix. Accordingly, any functional consequences in hybrids would more plausibly stem from changes in the gRNA repertoire than from maxicircle polymorphisms. Although precise edited mRNA sequences require direct transcript sequencing, only two parental polymorphisms fall within editing-requiring regions: (i) an A insertion at 2281 in P1 corresponds to one additional U in the pre-edited ND9 mRNA (because ND9 is encoded on the reverse strand), and (ii) a variant at position 3945 adds one U in COIII from P2. Given canonical gRNA targeting, both contexts are expected to yield identical edited products; thus, a functional impact on ND9 or COIII is unlikely. More broadly, stable biparental minicircle mixing can buffer editing against local loss of specific gRNA classes, potentially restoring editing capacity where needed [41].”
Comment:
- The article proposes the hypothesis that maxicircles replicate in the hybrid, but only the DNA from one of the parental strains remains. Since recombination has already been demonstrated in brucei(https://doi.org/10.1016/j.dnarep.2018.11.005), why is no recombination observed between parental maxicircles?
Response: We acknowledge that recombination can occur in kinetoplast DNA, particularly in the context of repair. However, there is no clear evidence for widespread inter-parental maxicircle recombinants in natural T. cruzi hybrids. Our data are consistent with rapid fixation of a single parental maxicircle without detectable mosaicism. In the mechanistic framework we propose, maxicircle subnetworks remain physically segregated and do not detach during replication, sharply reducing opportunities for inter-network strand exchange relative to minicircles. Together with effective copy-number dynamics and bottlenecks at network segregation, these features would favor unit-level inheritance over inter-parental recombination. We have clarified this rationale and now explicitly cite the evidence for recombination as a repair mechanism in T. brucei while explaining why our model predicts little to no inter-parental recombination for maxicircles in these T. cruzi hybrids.
The following sentence was added to the Disussion:
“Although recombination within kDNA has been demonstrated—particularly in T. brucei as part of repair pathways [40]—inter-parental maxicircle mosaicism was not observed in these hybrids because parental maxicircle subnetworks probably remain physically segregated.”
Comment:
- It would be interesting to discuss whether a higher amount of kDNA (maxi- and minicircles) could interfere with mRNA levels, RNA editing, and mitochondrial function.
- Why do hybrids lose kDNA after 800 generations?
Thank you for these suggestions. We agree and now note that higher kDNA dosage is unlikely to increase editing/respiration per se when parental mitochondria are functional. Experimental enlargements/mis-segregation of kDNA cause network instability, growth defects, and frequent kDNA loss [see references 46 and 48], indicating burden rather than benefit. Conversely, an expanded biparental minicircle repertoire can restore or buffer editing when specific gRNA classes have been lost or are limiting in parents. We have added a concise paragraph and references accordingly.
We have included this paragraph in discussion:
“Perturbations that enlarge or mis-segregate the kinetoplast (e.g., depletion of TAC components such as TAC102 or TAC40, or replication factors) produce oversized/unstable networks, growth impairment, and frequent kDNA loss—consistent with replication/segregation burden rather than increased editing or respiratory throughput [46, 48]. Conversely, an expanded, biparental minicircle repertoire can buffer or restore RNA editing if particular gRNA classes have been lost or are limiting in one or both parents . This framework is consistent with our observation that kDNA content declines over ~800 generations despite persistent biparental minicircle ancestry, suggesting selection against sustained kDNA enlargement with a potential short-term buffering benefit when gRNA classes are limiting.”
Minor points
Comment:
- In the legend of Figure 2 (lines 310–312), the colors representing the parental strains were swapped.
Response: Corrected.
Reviewer 2 Report
Comments and Suggestions for Authors
The manuscript by Nicolás Tomasini et al. is focused on the reanalysis of next genome sequencing data obtained from experimental hybrids of Trypanosoma cruzi to identify the most likely pattern of biparental inheritance of kinetoplastid DNA minicircles. The title and abstract reflect the essence of the study, and the authors' conclusions follow from the results. Although the authors conducted extensive bioinformatics analysis, including flow cytometry data, I believe the manuscript needs revision.
- The authors state in Section 2.6 that they analyzed flow cytometry data, but the Data Source section (lines 108-117) does not mention these data. Please add information about these data in this section.
- In my opinion, this work lacks broad readership; in particular, the content is created and described for a very narrow audience. To ensure this manuscript appeals to a wider audience, I strongly recommend expanding the Discussion section by adding an interpretation of the data in the context of the functionality of the findings.
- In the Introduction section, please include information about the rationale for experiments and analyses. Reviewers and other researchers reading this manuscript will benefit from some additional insight to fully appreciate its significance.
Author Response
Comment: The manuscript by Nicolás Tomasini et al. is focused on the reanalysis of next genome sequencing data obtained from experimental hybrids of Trypanosoma cruzi to identify the most likely pattern of biparental inheritance of kinetoplastid DNA minicircles. The title and abstract reflect the essence of the study, and the authors' conclusions follow from the results. Although the authors conducted extensive bioinformatics analysis, including flow cytometry data, I believe the manuscript needs revision.
- The authors state in Section 2.6 that they analyzed flow cytometry data, but the Data Source section (lines 108-117) does not mention these data. Please add information about these data in this section.
Response: We acknowledge the Reviewer for the comments. We have added a sentence in "Section 2.1 Data source" to clarify about flow cytometry data:
“Propidium iodide (PI)–based total DNA content profiles from epimastigote cultures of parental lines and hybrids at different at different time points were obtained from Matos et al. [26].”
Comment:
- In my opinion, this work lacks broad readership; in particular, the content is created and described for a very narrow audience. To ensure this manuscript appeals to a wider audience, I strongly recommend expanding the Discussion section by adding an interpretation of the data in the context of the functionality of the findings.
Response: We agree and have expanded the Discussion with a functional and evolutionary interpretation that links biparental minicircle mixing, maxicircle fixation, and transient dosage increases to mitochondrial performance, life-cycle plasticity, and epidemiological patterns:
“Biparental mixing of minicircles—documented in natural and successful hybrid DTUs such as TcV and TcVI [25]— reshaped the repertoire of guide RNAs [58], which in principle can buffer editing of maxicircle transcripts against loss of specific gRNA classes at the cost of a higher replication/segregation burden. This trade-off may also help explain why most laboratory hybrids show net losses of minicircles and maxicircles after ~800 generations while still maintaining mixed minicircle ancestry. Such dynamics may clarify why some hybrid lineages (e.g., TcV/TcVI) attain broad geographic ranges and epidemiological importance [59]. Practically, stable minicircle mixing explains why minicircle-based typing can retain biparental ancestry signals long after hybrid formation [25], whereas maxicircle markers read as uniparental which has relevance in epidemiological studies. Beyond kinetoplastids, biparental inheritance of mtDNA is uncommon with mechanisms to avoid it, and dissecting how kinetoplastids tolerate and even exploit biparental transmission can clarify the selective pressures that led most eukaryotes to evolve strict maternal inheritance, germline bottlenecks, and active paternal mtDNA elimination [60–62].”
Comment:
- In the Introduction section, please include information about the rationale for experiments and analyses. Reviewers and other researchers reading this manuscript will benefit from some additional insight to fully appreciate its significance.
Response: We appreciate this suggestion and have expanded the Introduction to articulate the rationale behind our experimental design and analyses.
The following paragraph in the introduction:
“In this study, we addressed gaps in the understanding of kDNA inheritance in T. cruzi by analyzing raw genome data from experimental TcI–TcI hybrids. We characterized the minicircle hypervariable region (mHVR) profiles across generations and quantified both minicircle and maxicircle abundance and persistence. This combined analysis supported a revised model of kDNA inheritance in hybrid trypanosomes.”
was replaced by:
“Although hybridization in trypanosomatids has been repeatedly observed and inferred, most conclusions about kinetoplast inheritance rely on maxicircle markers, which cannot reveal whether minicircles mix, replicate, or change in dosage after hybrid formation. Moreover, no prior study has jointly quantified minicircle and maxicircle copy numbers across many asexual generations to distinguish between models of kDNA inheritance. The experimental T. cruzi hybrids originally described by Gaunt and colleagues offer a unique opportunity: derived from known parents and propagated for ~800 generations [26], they enable reconstruction of minicircle repertoires and copy-number estimates. Reanalyzing these hybrids with a curated minicircle hypervariable region (mHVR) database and complementary PI-based cytometry constraints, we set out to (i) trace parental contributions to the minicircle pool over time, (ii) quantify total kDNA dosage for minicircles and maxicircles, and (iii) discriminate among alternative inheritance mechanisms and their predictions—thereby supporting a revised model of kDNA inheritance in hybrid trypanosomes.”
Round 2
Reviewer 2 Report
Comments and Suggestions for Authors
The authors have provided responses to all my comments therefore I do not have any further suggestions